# Inhibition of In Vitro Infection of Hepatitis B Virus by Human Breastmilk

**DOI:** 10.3390/nu14081561

**Published:** 2022-04-08

**Authors:** Yuqian Luo, Kuanhui Xiang, Jingli Liu, Ji Song, Jing Feng, Jie Chen, Yimin Dai, Yali Hu, Hui Zhuang, Yihua Zhou

**Affiliations:** 1Department of Laboratory Medicine, Nanjing Drum Tower Hospital and Jiangsu Key Laboratory for Molecular Medicine, Nanjing University Medical School, Nanjing 210008, China; yuqianluo31@foxmail.com (Y.L.); jingliliu1@163.com (J.L.); 2Department of Microbiology and Infectious Disease Center, School of Basic Medical Sciences, Peking University Health Science Center, Beijing 100191, China; xiangkuanhui174@126.com (K.X.); songji95@pku.edu.cn (J.S.); 3Department of Obstetrics and Gynecology, Nanjing Drum Tower Hospital, Nanjing University Medical School, Nanjing 210008, China; fengjingnyk@126.com (J.F.); cjnjmu2010@sina.com (J.C.); nj_daiyimin@126.com (Y.D.); 4Departments of Laboratory Medicine and Infectious Diseases, Nanjing Drum Tower Hospital, Nanjing University Medical School, Nanjing 210008, China

**Keywords:** human breastmilk, lactoferrin, HBsAg-binding protein, inhibition of infectivity of hepatitis B virus, mother-to-child transmission of HBV

## Abstract

Despite the presence of hepatitis B virus (HBV) in the human breastmilk of mothers infected with HBV, it has been shown that breastfeeding does not increase the risk of mother-to-child transmission (MTCT) of HBV. We tested the hypothesis that human breastmilk may contain active components that bind to HBV and inhibit the infectivity of HBV. The results show that human whey significantly inhibited the binding of the hepatitis B surface antigen (HBsAg) to its antibodies in competitive inhibition immunoassays. The far-western blotting showed that HBsAg bound to a protein of 80 kD in human whey, which was identified as lactoferrin by mass spectrometry. Competitive inhibition immunoassays further demonstrated that both human lactoferrin and bovine lactoferrin bound to HBsAg. Human whey, human lactoferrin, and bovine lactoferrin each significantly inhibited the infectivity of HBV in vitro. Our results indicate that human breastmilk can bind to HBsAg and inhibit the infectivity of HBV, and the active component is lactoferrin. The findings may explain the reason that breastfeeding has no additional risk for MTCT of HBV, although human breastmilk contains HBV. Our study provides experimental evidence that HBV-infected mothers should be encouraged to breastfeed their infants

## 1. Introduction

Infection of hepatitis B virus (HBV) is a worldwide health problem because about 250 million individuals globally are chronically infected with HBV and 20–40% of them will develop liver cirrhosis and hepatocellular carcinoma [1]. Mother-to-child transmission (MTCT) of HBV is a major cause of chronic HBV infection [2].

Human breastmilk provides various nutrients for infants. However, breastfeeding is a potential risk factor for the transmission of microorganisms. Mothers with the positive hepatitis B surface antigen (HBsAg) in circulation are also positive for HBsAg and HBV DNA in their breastmilk [3,4,5,6]. Moreover, breastmilk is easily contaminated by maternal HBV-containing blood, as cracks or fissures around the nipple inevitably occur frequently due to the inadequate breastfeeding management of mothers. Historically, breastfeeding was assumed to be capable of transmitting HBV to offspring [7]. However, even before the availability of immunoprophylaxis against HBV, studies showed that breastfeeding poses no additional risk for MTCT of HBV [8,9]. The underlying mechanism remains unknown.

Human breastmilk is known to contain biologically active components that have multiple functions [10]. Infants who are breastfed have a lower incidence of respiratory and gastrointestinal infections as well as malaria [11], and human breastmilk has the ability to inhibit viral infections [12,13,14]. Breastmilk contains a variety of antimicrobial substances such as immunoglobulins, lactoferrin, lysozyme, lactoperoxidase, mucins, fatty acids, hormones, and cytokines, which are relatively resistant against intestinal proteolysis and can function to both safeguard the lactating mammary gland and provide protection to the suckling infant at a time when its immune system is still immature [15]. In breastmilk, lactoferrin (LF) is particularly known as a potent inhibitor of several enveloped and naked viruses, such as rotavirus, enterovirus, and adenovirus [16,17]. The capability of LF to retain iron at acid pH, which characterizes infection and inflammation sites, together with its cationic nature, may be responsible for its ability to bind to various microbial and viral negative surface structures and to anionic molecules such as DNA [18,19]. Also, there is evidence that it can bind to at least some of the receptors used by coronaviruses and thereby block their entry [17]. Accordingly, a remarkable inhibiting effect of breastmilk on viral infection and replication of severe acute respiratory syndrome coronavirus 2 (SARS-CoV-2) and pangolin coronavirus has been demonstrated [20,21].

Considering the multiple antiviral effects of breastmilk, we hypothesized that human breastmilk may contain certain active components that bind to HBV and inhibit the infectivity of HBV. In the present study, we provide evidence that human breastmilk and lactoferrin bind to HBsAg and inhibit the infectivity of HBV in cell culture.

## 2. Materials and Methods

### 2.1. Milk and Serum Samples and Preparation of Whey Fractions

Human breastmilk was collected via pumps into sterile containers after disinfecting nipples with 75% ethanol. Whey fraction was obtained by centrifugation at 4000× *g* at 4 °C for 15 min and passed through 0.22 μm filters. All breastmilk samples were negative for anti-HBs. Bovine and goat whey were similarly collected. The protein concentrations were determined using DC protein assay (BIO-RAD, Hercules, CA, USA).

Serum samples were derived from HBV-infected patients with positive HBsAg and stored at −30 °C. All study subjects were informed and consenting. This study was approved by the institutional review boards of Nanjing Drum Tower Hospital (No. 2016-185).

### 2.2. Immunoassays for HBsAg and Hepatitis B e Antigen (HBeAg)

The semi-quantification of HBsAg and HBeAg was performed using enzyme-linked immunosorbent assays (ELISA) (Kehua Biotech, Shanghai, China). The quantification of HBsAg was performed by a microparticle enzyme immunoassay (Architect System, Abbott, North Chicago, IL, USA); the detection range of HBsAg was from 0.05–250 IU/mL. The yeast-expressed recombinant HBsAg was provided by the Tiantan Biological Corp (Beijing, China).

### 2.3. Competitive Inhibition Immunoassay for HBsAg-Binding Activity

In competitive HBsAg-binding inhibition immunoassays, tested competitors, including human whey, bovine whey, goat whey, human lactoferrin, recombinant human lactoferrin, and bovine lactoferrin (Sigma-Aldrich, St. Louis, MO, USA), were each pre-incubated with 0.04 μg/mL recombinant HBsAg or 20 IU/mL HBsAg-positive serum at room temperature for 5 min. The mixtures were subjected to semi-quantification or quantification of HBsAg. The percentage of decreased HBsAg titers after the pre-incubation was calculated as the HBsAg-binding inhibition ratio.

### 2.4. Quantitative Assay of HBV DNA

HBV DNA was quantified by fluorescent real-time PCR assays (Aikang Biotechnology, Hangzhou, China). Briefly, HBV DNA was extracted using the commercial kit from 20 μL samples. The PCR was run for 40 cycles and the fluorescence signal of the amplicons was detected on a StepOnePlus Real-Time PCR System (Applied Biosystems, Foster City, CA, USA). The linear detection range of this assay was 10^2^ to 10^8^ IU/mL.

### 2.5. Far-Western Blotting

Twenty μg whey proteins or 1 μg lactoferrin were separated by sodium dodecyl sulfate-polyacrylamide gel electrophoresis (SDS-PAGE) and transferred to polyvinylidene fluoride (PVDF) membranes. The membrane was washed with phosphate buffered saline (PBS) with 0.1% Tween 20 (PBST), blocked in 2% bovine albumin in PBST for 1 h, and incubated with 0.04 μg/mL purified yeast-expressed recombinant HBsAg in PBST for 30 min. Another PVDF membrane with the same protein lanes was incubated in PBST without HBsAg to serve as a negative control. After washing, membranes were incubated with horseradish peroxidase (HRP)-labeled anti-HBs (Kehua Biotech). The membranes were visualized using enhanced chemiluminescence reagents (Wako Pure Chemical, Osaka, Japan), followed by detection using a CCD-camera (Tanon, Shanghai, China).

### 2.6. Mass Spectrometry Analysis

The SDS-PAGE gels in far-western blotting corresponding to the HBsAg-bound proteins were subjected to mass spectrometry analysis using a TripleTOF 5600-plus system (AB Sciex, Foster City, CA, USA) coupled with an Ekspert NanoLC 425 (AB Sciex). ProteinPilot (AB Sciex) was used to process the acquired time-of-flight mass spectra data for peptide and protein identification.

### 2.7. Generation of HepG2-NTCP Cells

The HepG2-NTCP cells were prepared as previously reported [22]. Human NTCP cDNA was cloned into the neomycin resistance co-expressing lentiviral vector, pWPI-neo. HepG2 cells were transduced with NTCP-expressing lentiviruses and selected with 500 μg/mL G418. Once NTCP expression was confirmed by immunofluorescence staining, several single-cell clones were sub-cultured and tested for their permissiveness to HBV. One clone with the highest HBV-infection efficiency was used throughout this study.

### 2.8. In Vitro HBV Culture and Inhibition of HBV Infection

HBV stocks for cell cultures were prepared from the culture supernatants of HepG2.2.15 cells as previously reported [22]. Briefly, HepG2.2.15 cells were cultured in Dulbecco’s modified minimal essential medium (DMEM) supplemented with 10% FBS (ThermoFisher, Waltham, MA, USA). The supernatants were collected every other day, followed by centrifugation at 1000× *g* at 4 °C. The pooled supernatants were concentrated 100-fold via centrifugation using Centricon Plus-70 centrifugal filter devices (Millipore-Sigma, Billerica, MA, USA) to obtain the concentrated virus stock.

HepG2-NTCP cells were seeded in 96-well collagen-coated plates in DMEM supplemented with 10% FBS. The medium was changed to DMEM supplemented with 3% FBS and 2% DMSO (Sigma-Aldrich) on the next day. The cells were infected with the concentrated HBV stock at a multiplicity of infection (MOI) of 50. The stock was diluted in inoculum media, which is DMEM supplemented with 3% FBS, 2% DMSO, and 4% polyethylene glycol (PEG) 8000 (Sigma Aldrich). The inoculum volume was adjusted to the plate format (50 μL for 96-well plates). In the neutralization experiments, human/bovine/goat whey or natural human/recombinant human/bovine lactoferrin were each mixed with the stock virus, followed by inoculating the mixtures in the cultures of HepG2-NTCP cells. After 24 h, the inoculum was removed, cells were washed six times, and fresh medium was added. The supernatants were collected at 7 days after infection.

### 2.9. Statistical Analysis

Statistical analyses were conducted using GraphPad Prism 8 software (GraphPad Software Inc., San Diego, CA, USA). Values are shown as the mean of triplicates. Comparisons between the two groups were analyzed using the Student’s *t*-test. Comparisons between more than two groups were analyzed using the ordinary one-way ANOVA followed by Dunnet’s post hoc test. Values of *p* < 0.05 were considered statistically significant. All authors had access to the study data and reviewed and approved the final manuscript.

## 3. Results

### 3.1. Inhibition of Binding of HBsAg to Anti-HBs by Human Breastmilk

HBsAg is the major protein on HBV virions. As mentioned before, breastmilk contains components that are known to bind with viral proteins [18,19]. We speculated that human breastmilk components may also bind to HBV. Thus, we first investigated whether human breastmilk may bind to HBsAg by a competitive inhibition immunoassay. In the assay, we incubated the purified recombinant HBsAg or serum HBsAg with PBS or various concentrations (2, 5, 10 mg/mL) of human, bovine, and goat wheys, respectively. The concentration of recombinant or serum HBsAg was determined based on the readings within the range of the calibration curve. The residual HBsAg titers of the mixtures were then quantified. The results showed that human whey significantly (*p* < 0.0001) inhibited the binding of both recombinant HBsAg (Figure 1A) and serum HBsAg (Figure 1B) to anti-HBs, and the inhibition efficiency was as high as 80–90% when 10 mg/mL human whey was used (Figure 1A,B), whereas bovine whey at 10 mg/mL showed limited inhibiting activity (Figure 1A,B).

To exclude the possibility of binding of human whey to anti-HBs coated on the ELISA plate, we conducted another competitive inhibition immunoassay in which anti-HBs coated microplates were added with PBS or various concentrations (2, 5, 10 mg/mL) of human wheys, respectively. After washing, the plate was used to detect recombinant HBsAg or human serum HBsAg. The results showed that human whey did not bind to anti-HBs (Appendix A).

### 3.2. Identification of Lactoferrin as a Component in Human Whey to Bind to HBsAg

Since the above results indicate that components of human whey may bind to HBsAg, we performed far-western blotting to investigate such components. Human whey proteins were separated by SDS-PAGE and transferred to a PVDF membrane, followed by incubation with recombinant HBsAg. The membrane was probed by HRP-conjugated anti-HBs. A band at 80 kD was visible in the lanes loaded with human wheys, but the lanes loaded with bovine or goat wheys showed hardly visible bands (Figure 1C). Meanwhile, the PVDF membrane without HBsAg incubation did not show any band (Figure 1C).

To identify the protein component(s) that bound to HBsAg in Figure 1C, we separated the human wheys in SDS-PAGE and cut the gel at 80 kD for mass spectrometry analysis (Appendix A). The results showed a high-scored protein, with 93.4% amino acid residues identical to the human lactoferrin (Appendix A). Thus, we assumed human lactoferrin as an HBsAg-binding protein in human whey.

To verify the binding of lactoferrin with HBsAg, we performed competitive inhibition immunoassays after direct incubation of recombinant HBsAg with human lactoferrin, recombinant human lactoferrin, and bovine lactoferrin, respectively. The results showed that each lactoferrin efficiently inhibited the binding of HBsAg (Figure 2A). Far-western blotting further confirmed the binding of HBsAg with each lactoferrin based on the visible bands at 80 kD (Figure 2B).

To further investigate whether lactoferrin is the only component in human whey to bind to HBsAg, we incubated human whey with antibodies against lactoferrin and then measured the inhibition of the mixture on the binding of HBsAg to anti-HBs in the competitive inhibition immunoassay. The results showed that in the presence of antibodies against lactoferrin, the inhibition efficiency on the binding of HBsAg to anti-HBs of human whey was decreased by approximately 75% (Figure 2C).

### 3.3. Inhibition of HBV Infectivity in HepG2-NTCP Cells by Human Whey

Having demonstrated the binding of human whey to HBsAg, we attempted to clarify whether human whey inhibits the infectivity of HBV. We performed the HBV infection experiments based on a recently developed HBV cell culture system, the HepG2-NTCP cells [22]. We used human wheys from mothers with either negative or positive HBsAg, the bovine and goat whey, PBS (mock), and HBIG (1000 mIU/mL), respectively (Figure 3). We separately inoculated the mixtures of concentrated HBV and various concentrations of human wheys or controls in HepG2-NTCP cells. The culture supernatants were measured for HBeAg and HBV DNA levels to determine the HBV infectivity. In the mock group, high levels of HBeAg were detected (Figure 3A,B). HBIG completely abolished HBV infection, as HBeAg was not detectable (Figure 3). The supernatants of cultures in the presence of human whey from mothers with either negative (Figure 3A) or positive (Figure 3B) HBsAg had significantly (*p* < 0.05) reduced levels of HBeAg (Figure 3A,B). Dilution of human whey showed that the inhibition of human whey on HBV infectivity was concentration-dependent (Figure 3A,B). Moreover, HBV DNA levels in the supernatant of cell cultures in the presence of human whey were significantly (*p* < 0.0001) reduced (Figure 3C,D). In addition, the inhibition of HBV infectivity was observed only in the presence of human whey but was not found in the presence of bovine or goat whey (Figure 3E,F).

To exclude the possibility that human whey might affect cell growth and viability to reduce the synthesis of HBeAg and HBV DNA, we analyzed the viability of cultured cells incubated with human whey at 5 mg/mL. The results showed no significant difference in the number of viable cells treated with or without human whey (Appendix A).

### 3.4. Inhibition of HBV Infectivity in HepG2-NTCP Cells by Human and Bovine Lactoferrin

As lactoferrin was identified to bind to HBsAg (Figure 1 and Figure 2), we further tested the inhibition of HBV infectivity by human lactoferrin and bovine lactoferrin. We thus inoculated HepG2-NTCP cells with a mixture of HBV and these three types of lactoferrin at various concentrations (0.04, 0.2, 1 mg/mL) for 24 h, and evaluated the HBV infectivity by measuring the levels of HBeAg and HBV DNA in the supernatants of cell cultures at 7 days after infection. All three sources of lactoferrin significantly (*p* < 0.0001) inhibited the infectivity of HBV based on the levels of HBeAg and HBV DNA in the culture supernatant (Figure 4).

## 4. Discussion

In the present study, we demonstrated that human breastmilk binds to HBsAg and inhibits the infectivity of HBV in the cell culture and identified that lactoferrin in human breastmilk is the active component for the HBsAg binding and the inhibition of infectivity of HBV. The results may explain that the breastmilk of HBV-infected mothers contains the viruses, but breastfeeding does not cause MTCT of HBV.

We showed that human breastmilk efficiently inhibited the binding of HBsAg to anti-HBs in the competitive inhibition immunoassays (Figure 1). The inhibition should be achieved by binding to HBsAg rather than binding to anti-HBs, since pre-incubation of human breastmilk in an anti-HBs-coated plate did not interfere with the binding to HBsAg (Appendix A) and HRP-conjugated anti-HBs did not react with any of human breastmilk components (Figure 1C and Figure 2B). The counteraction of inhibition against lactoferrin by antibodies (Figure 2C) further demonstrated the inhibitory effect of lactoferrin in human whey. Thus, these results demonstrate that human breastmilk binds to HBsAg and the main active component for the binding is lactoferrin. As HBsAg constitutes the major protein on the envelope of HBV, it is reasonable to consider that human breastmilk can bind to HBV through the binding of lactoferrin to HBsAg.

The binding of human breastmilk to HBsAg encouraged us to further study whether the binding sequentially affects the infectivity of HBV. Using HepG2-NTCP cells, we showed that infectivity of HBV was inhibited by human breastmilk as well as human and bovine lactoferrin, as the levels of HBeAg and HBV DNA in the supernatants of the cultures in the presence of human whey, human lactoferrin, or bovine lactoferrin were significantly decreased (Figure 3 and Figure 4).

In HepG2-NTCP cells, bovine lactoferrin inhibited the infectivity of HBV (Figure 4E,F), which is in agreement with the results that bovine lactoferrin bound to HBsAg in the far-western blotting (Figure 2B). However, in the competitive inhibition assays, bovine or goat whey had little inhibition on the binding of HBsAg to anti-HBs (Figure 1A,B) and HBsAg did not show obvious reactivity to bovine or goat whey proteins on the membrane separated by SDS-PAGE (Figure 1C). It is likely that bovine or goat whey contains much lower concentrations of lactoferrin [16,23,24].

It has been recognized that lactoferrin has multiple antimicrobial activities [16]. Recently, we demonstrated a remarkable inhibitory effect of breastmilk on viral infection and replication of severe acute respiratory syndrome coronavirus 2 [20,21]. Previously, some scholars reported that human and bovine lactoferrin or lactoferrin-derived peptide inhibited HBV infection in human hepatocytes PH5CH8 and HepaRG cells [25,26]. Hara et al. found that pre-incubation of HBV and lactoferrin did not inhibit the replication of HBV, but pre-incubation of lactoferrin and hepatocytes inhibited the viral replication, and thus proposed that lactoferrin inhibits the viral replication through binding to hepatocytes, rather than binding to HBV [25]. In the present study, we verified the bindings of human breastmilk as well as lactoferrin to HBsAg, which had not been reported before. We further demonstrated the inhibition of human breastmilk and lactoferrin on the infectivity of HBV in recently developed HepG2-NTCP cell cultures. In HepG2-NTCP cells, HBV infection can be established at a much lower MOI compared to that in the previous studies [25,26,27], thus increasing the sensitivity and resolution to test any potential antiviral substance.

The demonstration of the inhibition ability of human breastmilk on HBV infectivity may help us to understand the underlying logic that breastfeeding has no additional risk for MTCT of HBV in infants born to HBV-infected mothers, which has been observed in numerous studies before and after the availability of HBIG and the hepatitis B vaccine [7,8,9,28]. The guidelines also recommend breastfeeding infants born to HBV-infected mothers [29,30,31,32,33]. However, because the breastmilk of HBV-infected mothers contains HBV [4,5], a considerable proportion of physicians and parents still believe that breastfeeding may transmit HBV [34,35], leading to the persistently low breastfeeding rate in infants of HBV-infected mothers [36]. Thus, the evidence that shows human breastmilk can inhibit the infectivity of HBV attained in the present study can enhance the confidence of physicians and parents, encouraging them to accept the recommendation that breastfeeding is not contraindicated for infants born to HBV-infected mothers.

## 5. Conclusions

We demonstrated that human breastmilk could bind to HBsAg and inhibit the infectivity of HBV in HepG2-NTCP cells, and the active component is lactoferrin. The results indicate that lactoferrin may act as a protective factor against HBV infection during breastfeeding.

## Figures and Tables

**Figure 1 nutrients-14-01561-f001:**
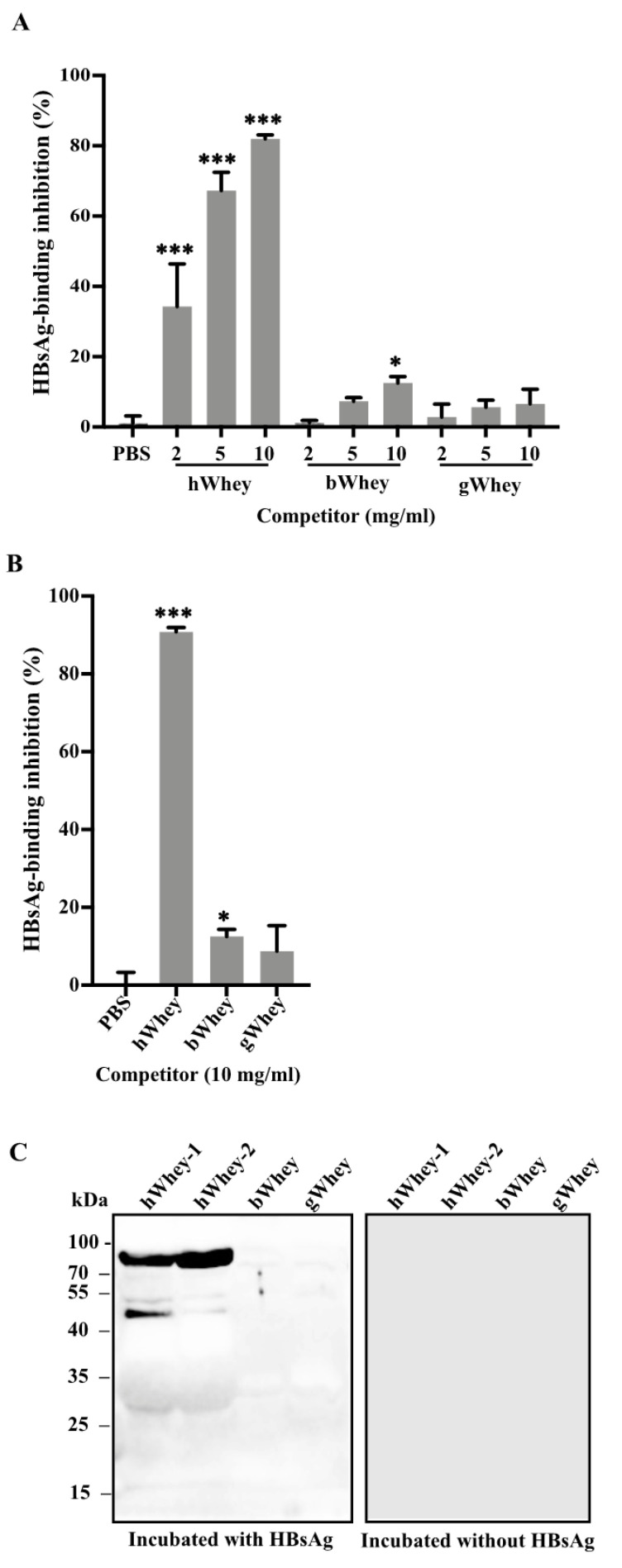
Human whey binds with HBsAg. (**A**,**B**) Inhibition of binding of HBsAg to anti-HBs by human breast milk tested by competition immunoassays. Recombinant HBsAg (0.04 μg/mL) (**A**) and human serum HBsAg (20 IU/mL) (**B**) were each pre-incubated with human whey (hWhey), bovine whey (bWhey), and goat whey (gWhey) for 5 min, respectively. PBS served as a negative control. The residual HBsAg titers in each mixture were then measured using ELISA (A) and Abbott chemiluminescent microparticle immunoassays (**B**). The percentage of decreased HBsAg titers after the pre-incubation was calculated as HBsAg-binding inhibition ratio. (**C**) Human whey from two healthy individuals, bovine whey, and goat whey were separated by SDS-PAGE (20 μg total protein per lane) and transferred to PVDF membrane, followed by the incubation with (left) or without (right) recombinant HBsAg (0.04 μg/mL). After washing, the membrane was probed by HRP-conjugated anti-HBs. A typical result showed that a band at approximately 80 kD was detected in the lanes loaded with human whey. The PVDF membrane without the incubation with HBsAg did not show any band. Data are presented as means ± SD. *: *p* < 0.05 and ***: *p* < 0.0001, compared with PBS control group.

**Figure 2 nutrients-14-01561-f002:**
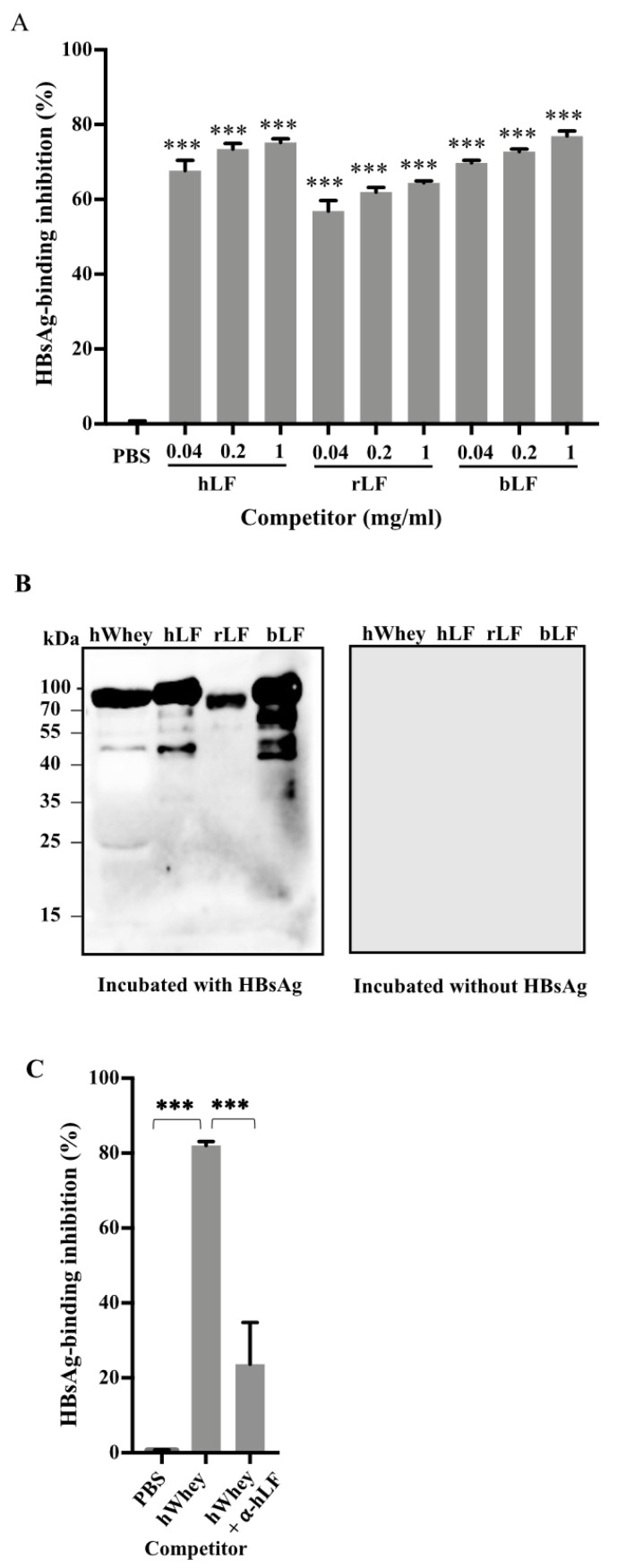
Human lactoferrin binds with HBsAg. (**A**) Recombinant HBsAg (0.04 μg/mL) was pre-incubated with natural human lactoferrin (hLF), recombinant human lactoferrin (rLF), and natural bovine lactoferrin (bLF), respectively. PBS served as a negative control. The residual HBsAg titers were then measured using ELISA. (**B**) Natural hLF, rLF, and bLF were separated by SDS-PAGE (2 μg total protein per lane) and transferred to a PVDF membrane, followed by the incubation with (left) or without (right) recombinant HBsAg (0.04 μg/mL). After washing, the membrane was probed by HRP-conjugated anti-HBs. A typical result showed that a band at approximately 80 kD was detected in all lanes loaded with lactoferrins. The PVDF membrane without the incubation with HBsAg did not show any band. (**C**) Human whey (hWhey) at 10 mg/mL was pre-incubated with or without 10 μg/mL anti-human lactoferrin (α-hLF), followed by the incubation with 0.04 μg/mL recombinant HBsAg or 20 IU/mL human serum HBsAg. The HBsAg-binding inhibition ratio was calculated as described in Figure 1. Data are presented as means ± SD. ***: *p* < 0.0001, compared with PBS control group.

**Figure 3 nutrients-14-01561-f003:**
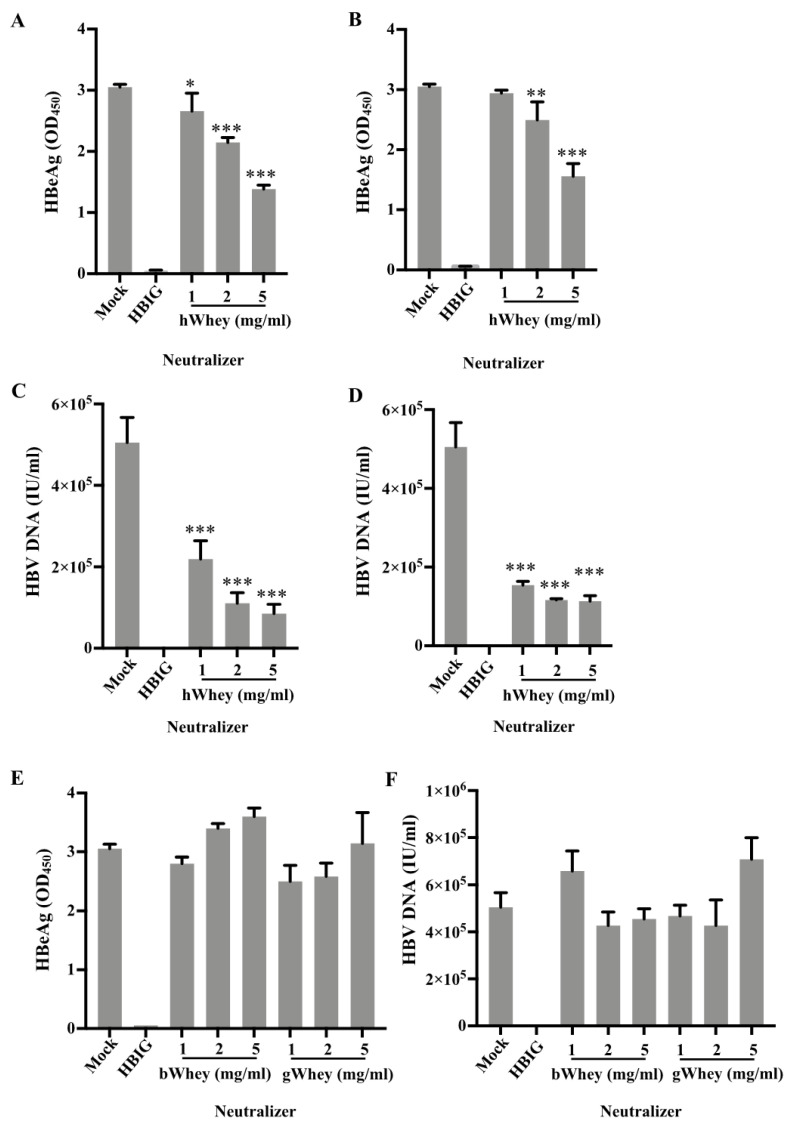
Human whey inhibits HBV infectivity in HepG2-NTCP cells. HepG2-NTCP cells were inoculated with a mixture of concentrated HBV (50 GEQ/cell) and (**A**–**D**) human whey (hWhey), (**E**,**F**) bovine whey (bWhey), or goat whey (gWhey) for 24 h. PBS and HBIG (1000 mIU/mL) served as mock and positive controls, respectively. Anti-HBs negative human whey fractions were derived from individuals with negative HBV markers (**A**,**C**) or from individuals with positive HBsAg and negative anti-HBs (**B**,**D**). The inoculation mixtures were removed at 24 h, and cells were washed with PBS six times and kept for up to 7 days in infection media (details in Materials and Methods). The culture supernatants were measured for secreted HBeAg (**A**,**B**,**E**) and HBV DNA (**C**,**D**,**F**) to evaluate the HBV infectivity. Data are presented as means ± SD. *: *p* < 0.05, **: *p* < 0.01, and ***: *p* < 0.0001, compared with PBS control group.

**Figure 4 nutrients-14-01561-f004:**
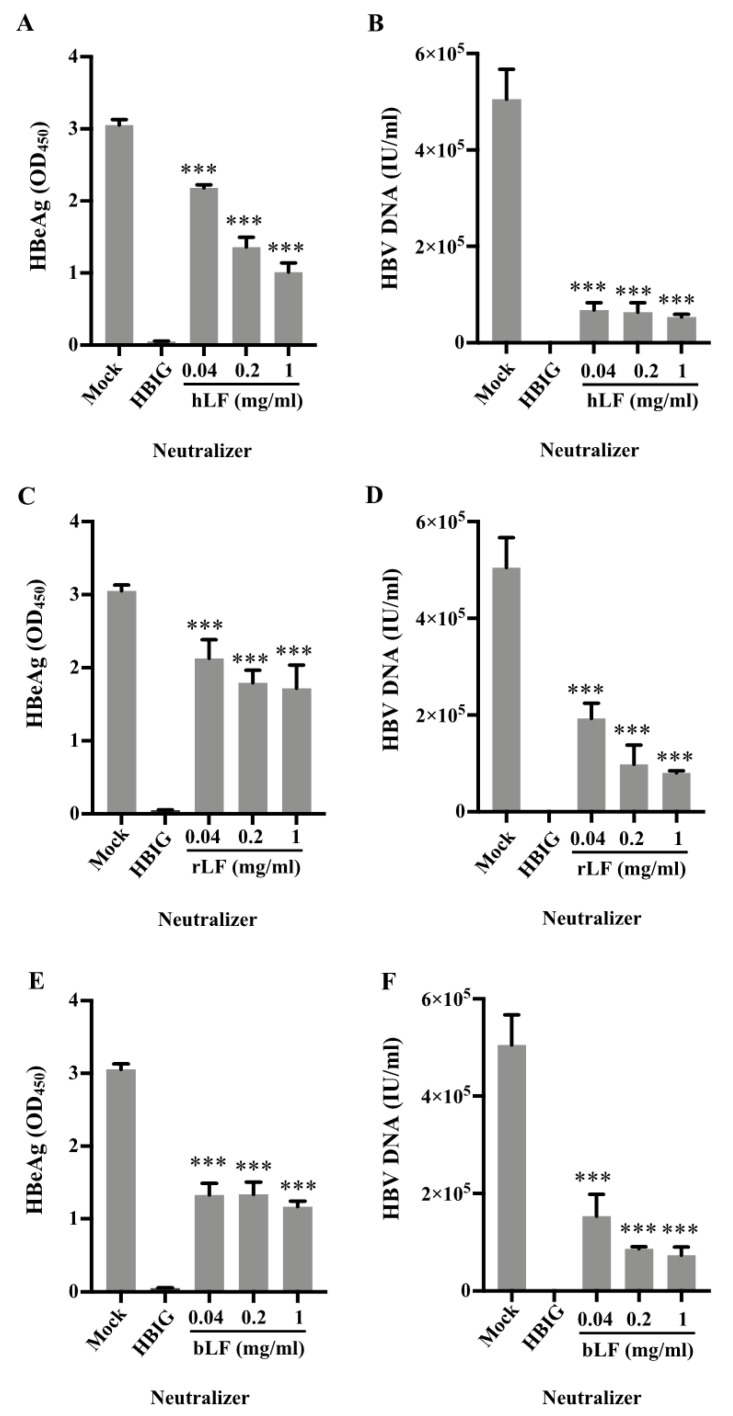
Natural human lactoferrin, recombinant human lactoferrin, and bovine lactoferrin each inhibits HBV infectivity in HepG2-NTCP cells. HepG2-NTCP cells were inoculated with a mixture of concentrated HBV (50 GEQ/cell) and natural human lactoferrin (hLF) (**A**,**D**), recombinant human lactoferrin (rLF) (**B**,**E**), or natural bovine lactoferrin (bLF) (**C**,**F**) for 24 h. PBS and HBIG (1000 mIU/mL) served as mock and positive controls, respectively. The inoculation mixtures were removed at 24 h, and cells were thoroughly washed with PBS six times and kept for up to 7 days in infection media (details in Materials and Methods). The culture supernatants were measured for secreted HBeAg (**A**,**C**,**E**) and HBV DNA (**B**,**D**,**F**) to evaluate the HBV infectivity. Data are presented as means ± SD. ***: *p* < 0.0001, compared with the PBS control group.

## Data Availability

Data, analytic, methods, materials, and study materials will be made available upon request.

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
