# Peer review of "Inhibition of In Vitro Infection of Hepatitis B Virus by Human Breastmilk"

_nutrients, 2022, doi:10.3390/nu14081561_

Round 1

Reviewer 1 Report

This was an interesting and well thought out study describing the mechanism by which breastmilk may protect against infection in infants fed by HBsAg positive mothers.  The study was systematic in demonstrating first binding capacity, second isolating the protein found responsible for binding and third examining effect on infectivity.   This is of interest to persons specializing among others in pediatrics, infectious disease, public health and nutrition 

Author Response

We thank the reviewer for this kind comment. We have improved our English writing thoroughly. 

Reviewer 2 Report

This is a thorough investigation of the way in which breastmilk inhibits the infectivity of HBV and should reassure health professionals and mothers that it continues to be safe for HBV-positive mothers to breastfeed their babies.

Line 44 ‘…cracks or fissures around the nipple inevitably occur during breastfeeding.’

Cracks and fissures are not inevitable – they occur because the breastfeeding management of mothers is inadequate and leads to problems.

Line 48 ‘The underlying mechanism remains unknown.’

There are many studies that show antiviral activity of breastmilk and components with antiviral activity have been identified, including lactoferrin.

Kell, D. B., Heyden, E. L., & Pretorius, E. (2020). The biology of lactoferrin, an iron-binding protein that can help defend against viruses and bacteria. Frontiers in Immunology, 1221. https://doi.org/10.3389/fimmu.2020.01221

The Introduction should include better information about:

  • the known antiviral properties of breastmilk, the current information Lines 50 to 51 is inadequate and
  • the known antiviral properties of breastmilk lactoferrin – there is no information presented

Line 147- 149 ‘As we speculated that human breastmilk components may bind to HBV, we first investigated whether human breastmilk may bind to HBsAg by a competitive inhibition immunoassay.’

Evidence for this speculation needs to be clearly provided in the Introduction and the aims of the research, which actually aren’t addressed in the Introduction.

Author Response

Line 44 ‘…cracks or fissures around the nipple inevitably occur during breastfeeding.’

Cracks and fissures are not inevitable – they occur because the breastfeeding management of mothers is inadequate and leads to problems.

Answer: We revised the manuscript accordingly (lines 44-45).

Line 48 ‘The underlying mechanism remains unknown.’

There are many studies that show antiviral activity of breastmilk and components with antiviral activity have been identified, including lactoferrin.

Kell, D. B., Heyden, E. L., & Pretorius, E. (2020). The biology of lactoferrin, an iron-binding protein that can help defend against viruses and bacteria. Frontiers in Immunology, 1221. https://doi.org/10.3389/fimmu.2020.01221

Answer: We added more content regarding the antiviral activity of breastmilk and lactoferrin and cited the suggested paper (lines 56-73)

The Introduction should include better information about:

  • the known antiviral properties of breastmilk, the current information Lines 50 to 51 is inadequate and
  • the known antiviral properties of breastmilk lactoferrin – there is no information presented

Answer: We included the suggested points in the introduction as suggested (lines 56-73).

Line 147- 149 ‘As we speculated that human breastmilk components may bind to HBV, we first investigated whether human breastmilk may bind to HBsAg by a competitive inhibition immunoassay.’

Evidence for this speculation needs to be clearly provided in the Introduction and the aims of the research, which actually aren’t addressed in the Introduction.

Answer: We provided the basis of this speculation in the introduction and right before this sentence (lines 56-73, 182-183)